# AN ACTOR-CRITIC ALGORITHM FOR LEARNING RATE LEARNING

**Chang Xu**
Nankai University
`changxu@nbjl.nankai.edu.cn`

**Tao Qin**
Microsoft Research Asia
`taoqin@microsoft.com`

**Gang Wang**
Nankai University
`wgzwp@nbjl.nankai.edu.cn`

**Tie-Yan Liu**
Microsoft Research Asia
`tie-yan.liu@microsoft.com`

## ABSTRACT

Stochastic gradient descent (SGD), which updates the model parameters by adding a local gradient times a learning rate at each step, is widely used in model training of machine learning algorithms such as neural networks. It is observed that the models trained by SGD are sensitive to learning rates and good learning rates are problem specific. To avoid manually searching of learning rates, which is tedious and inefficient, we propose an algorithm to automatically learn learning rates using actor-critic methods from reinforcement learning (RL). In particular, we train a policy network called actor to decide the learning rate at each step during training, and a value network called critic to give feedback about quality of the decision (e.g., the goodness of the learning rate outputted by the actor) that the actor made. Experiments show that our method leads to good convergence of SGD and can prevent overfitting to a certain extent, resulting in better performance than human-designed competitors.

## 1 INTRODUCTION

While facing large scale of training data, stochastic learning such as stochastic gradient descent (SGD) is usually much faster than batch learning and often results in better models. An observation for SGD methods is that their performances are highly sensitive to the choice of learning rate LeCun et al. (2012). Clearly, setting a static learning rate for the whole training process is insufficient, since intuitively the learning rate should decrease when the model becomes more and more close to a (local) optimum as the training goes on over time Maclaurin et al. (2015). Although there are some empirical suggestions to guide how to adjust the learning rate over time in training, it is still a difficult task to find a good policy to adjust the learning rate, given that good policies are problem specific and depend on implementation details of a machine learning algorithm. One usually needs to try many times and adjust the learning rate manually to accumulate knowledge about the problem. However, human involvement often needs domain knowledge about the target problems, which is inefficient and difficult to scale up to different problems. Thus, a natural question arises: can we automatically adjust the learning rate? This is exactly the focus of this work and we aim to automatically learn the learning rates for SGD based machine learning (ML) algorithms without human-designed rules or hand-crafted features.

By examining the current practice of learning rate control/adjustment, we have two observations. First, learning rate control is a sequential decision process. At the beginning, we set an initial learning rate. Then at each step, we decide whether to change the learning rate and how to change it, based on the current model and loss, training data at hand, and maybe history of the training process. As suggested in Orr & Müller (2003), one well-principled method for estimating the ideal learning rate that is to decrease the learning rate when the weight vector oscillates, and increase it when the weight vector follows a relatively steady direction. Second, although at each step some immediate reward (e.g., the loss decrement) can be obtained by taking actions, we care more about the performance of the final model found by the ML algorithm. Consider two different learning rate

control policies: the first one leads to fast loss decrease at the beginning but gets saturated and stuck in a local minimum quickly, while the second one starts with slower loss decrease but results in much smaller final loss. Obviously, the second policy is better. That is, we prefer long-term rewards over short-term rewards.

Combining the two observations, it is easy to see that the problem of finding a good policy to control/adjust learning rate falls into the scope of reinforcement learning (RL) Sutton & Barto (1998), if one is familiar with RL. Inspired by the recent success of RL for sequential decision problems, in this work, we leverage RL techniques and try to learn the learning rate for SGD based methods.

We propose an algorithm to learn the learning rate within the actor-critic framework Sutton (1984); Sutton et al. (1999); Barto et al. (1983); Silver et al. (2014) from RL. In particular, an actor network is trained to take an action that decides the learning rate for current step, and a critic network is trained to give feedbacks to the actor network about long-term performance and help the actor network to adjust itself so as to perform better in the future steps. The main contributions of this paper include:

- We propose an actor-critic algorithm to automatically learn the learning rate for ML algorithms.
- Long-term rewards are exploited by the critic network in our algorithm to choose a better learning rate at each step.
- We propose to feed different training examples to the actor network and the critic network, which improve the generalization performance of the learnt ML model.
- A series of experiments validate the effectiveness of our proposed algorithm for learning rate control.

## 2 RELATED WORK

### 2.1 IMPROVED GRADIENT METHODS

Our focus is to improve gradient based ML algorithm through automatic learning of learning rate. Different approaches have been proposed to improve gradient methods, especially for deep neural networks.

Since SGD solely rely on a given example (or a mini-batch of examples) to compare gradient, its model update at each step tends to be unstable and it takes many steps to converge. To solve this problem, momentum SGD Jacobs (1988) is proposed to accelerate SGD by using recent gradients. RMSprop Tieleman & Hinton (2012) utilizes the magnitude of recent gradients to normalize the gradients. It always keeps a moving average over the root mean squared gradients, by which it divides the current gradient. Adagrad Duchi et al. (2011) adapts component-wise learning rates, and performs larger updates for infrequent and smaller updates for frequent parameters. Adadelta Zeiler (2012) extends Adagrad by reducing its aggressive, monotonically decreasing learning rate. Instead of accumulating all past squared gradients, Adadelta restricts the window of accumulated past gradients to some fixed size. Adam Kingma & Ba (2014) computes component-wise learning rates using the estimates of first and second moments of the gradients, which combines the advantages of AdaGrad and RMSProp.

Senior et al. (2013); Sutton (1992); Darken & Moody (1990) focus on predefining update rules to adjust learning rates during training. A limitation of these methods is that they have additional free parameters which need to be set manually. Another recent work Daniel et al. (2016) studies how to automatically select step sizes, but it still requires hand-tuned features. Schaul et al. (2013) proposes a method to choose good learning rate for SGD, which relies on the square norm of the expectation of the gradient, and the expectation of the square norm of the gradient. The method is much more constrained than ours and several assumption should be met.

### 2.2 REINFORCEMENT LEARNING

Since our proposed algorithm is based on RL techniques, here we give a very brief introduction to RL, which will ease the description of our algorithm in next section.

Reinforcement learning Sutton (1988) is concerned with how an agent acts in a stochastic environment by sequentially choosing actions over a sequence of time steps, in order to maximize a cumulative reward. In RL, a state $s^t$ encodes the agents observation about the environment at a time step $t$, and a policy function $\pi(s^t)$ determines how the agent behaves (e.g., which action to take) at state $s^t$. An action-value function (or, Q function) $Q_\pi(s^t, a^t)$ is usually used to denote the cumulative reward of taking action $a^t$ at state $s^t$ and then following policy $\pi$ afterwards.

Many RL algorithms have been proposed Sutton & Barto (1998); Watkins & Dayan (1992), and many RL algorithms Sutton (1984); Sutton et al. (1999); Barto et al. (1983); Silver et al. (2014) can be described under the actor-critic framework. An actor-critic algorithm learns the policy function and the value function simultaneously and interactively. The policy structure is known as the actor, and is used to select actions; the estimated value function is known as the critic, and it criticizes the actions made by the actor.

Recently, deep reinforcement learning, which uses deep neural networks to approximate/represent the policy function and/or the value function, have shown promise in various domains, including Atari games Mnih et al. (2015), Go Silver et al. (2016), machine translation Bahdanau et al. (2016), image recognition Xu et al. (2015), etc.

## 3 METHOD

In this section, we present an actor-critic algorithm that can automate the learning rate control for SGD based machine learning algorithms.

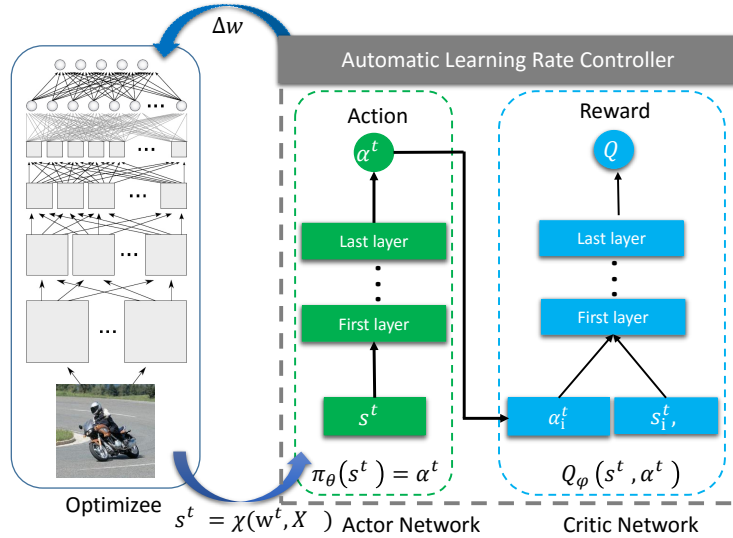

Figure 1: The framework of our proposed automatic learning rate controller.

Many machine learning tasks need to train a model with parameters $\omega$ by minimizing a loss function $f$ defined over a set $X$ of training examples:

$$\omega^* = \arg\min_\omega f_\omega(X). \tag{1}$$

A standard approach for the loss function minimization is gradient descent, which sequentially updates the parameters using gradients step by step:

$$\omega^{t+1} = \omega^t - a^t \nabla f^t, \tag{2}$$

where $a^t$ is the learning rate at step $t$, and $\nabla f^t$ is the local gradient of $f$ at $\omega^t$. Here one step can be the whole batch of all the training data, a mini batch of tens/hundreds of examples, or a random sample.

It is observed that the performance of SGD based methods is quite sensitive to the choice of $a^t$ for non-convex loss function $f$. Unfortunately, $f$ is usually non-convex with respect to the parameters

$w$ in many ML algorithms, especially for deep neural networks. We aim to learn a learning rate controller using RL techniques that can automatically control $a^t$.

Figure 1 illustrates our automatic learning rate controller, which adopts the actor-critic framework in RL. The basic idea is that at each step, given the current model $\omega^t$ and training sample $x$, an actor network is used to take an action (the learning rate $a^t$, and it will be used to update the model $\omega^t$), and a critic network is used to estimate the goodness of the action. The actor network will be updated using the estimated goodness of $a^t$, and the critic network will be updated by minimizing temporal difference (TD) Sutton & Barto (1990) error. We describe the details of our algorithm in the following subsections.

## 3.1 ACTOR NETWORK

The actor network, which is called policy network in RL, plays the key role in our algorithm: it determines the learning rate control policy for the primary ML algorithm[1] based on the current model, training data, and maybe historical information during the training process.

Note that $\omega^t$ could be of huge dimensions, e.g., one widely used image recognition model VGGNet Simonyan & Zisserman (2014) has more than 140 million parameters. If the actor network takes all of those parameters as the inputs, its computational complexity would dominate the complexity of the primary algorithm, which is unfordable. Therefore, we propose to use a function $\chi(\cdot)$ to process and yield a compact vector $s^t$ as the input of the actor network. Following the practice in RL, we call $\chi(\cdot)$ the state function, which takes $\omega^t$ and the training data $x$ as inputs:

$$s^t = \chi(\omega^t, X). \tag{3}$$

Then the actor network $\pi_\theta(\cdot)$ parameterized by $\theta$ yields an action $a^t$:

$$\pi_\theta(s^t) = a^t, \tag{4}$$

where the action $a^t \in \mathbb{R}$ is a continuous value. When $a^t$ is determined, we update the model of the primary algorithm by Equation 2.

Note that the actor network has its own parameters and we need to learn them to output a good action. To learn the actor network, we need to know how to evaluate the goodness of an actor network. The critic network exactly plays this role.

## 3.2 CRITIC NETWORK

Recall that our goal is to find a good policy for learning rate control to ensure that a good model can be learnt eventually by the primary ML algorithm. For this purpose, the actor network needs to output a good action $a^t$ at state $s^t$ so that finally a low training loss $f(\cdot)$ can be achieved. In RL, the Q function $Q_\pi(s, a)$ is often used to denote the long term reward of the state-action pair $s, a$ while following the policy $\pi$ to take future actions. In our problem, $Q_\pi(s^t, a^t)$ indicates the accumulative decrement of training loss starting from step $t$. We define the immediate reward at step $t$ as the one step loss decrement:

$$r^t = f^t - f^{t+1}. \tag{5}$$

The accumulative value $R_\pi^t$ of policy $\pi$ at step $t$ is the total discounted reward from step $t$:

$$R_\pi^t = \Sigma_{k=t}^{T} \gamma^{k-t} r(s^k, a^k),$$

where $\gamma \in (0, 1]$ is the discount factor.

Considering that both the states and actions are uncountable in our problem, the critic network uses a parametric function $Q_\varphi(s, a)$ with parameters $\varphi$ to approximate the Q value function $Q_\pi(s, a)$.

---

[1]Here we have two learning algorithms. We call the one with learning rate to adjust as the primary ML algorithm, and the other one which optimizes the learning rate of the primary one as the secondary ML algorithm.

### 3.3 Training of Actor and Critic Networks

The critic network has its own parameters $\varphi$, which is updated at each step using TD learning. More precisely, the critic is trained by minimizing the square error between the estimation $Q_\varphi(s^t, a^t)$ and the target $y^t$:

$$y^t = r^t + \gamma Q_\varphi(s^{t+1}, a^{t+1}). \tag{6}$$

The TD error is defined as:

$$\begin{aligned} \delta^t &= y^t - Q_\varphi(s^t, a^t) \\ &= r^t + \gamma Q_\varphi(s^{t+1}, \pi_\theta(s^{t+1})) - Q_\varphi(s^t, a^t) \end{aligned} \tag{7}$$

The weight update rule follows the on-policy deterministic actor-critic algorithm. The gradients of critic network are:

$$\nabla\varphi = \delta^t \nabla_\varphi Q_\varphi(s^t, a^t), \tag{8}$$

The policy parameters $\theta$ of the actor network is updated by ensuring that it can output the action with the largest Q value at state $s^t$, i.e., $a^* = \arg\max_a Q_\varphi(s^t, a)$. Mathematically,

$$\nabla\theta = \nabla_\theta \pi_\theta(s^{t+1}) \nabla_a Q_\varphi(s^{t+1}, a^{t+1})|_{a=\pi_\theta(s)}. \tag{9}$$

---

**Algorithm 1** Actor-Critic Algorithm for Learning Rate Learning

---

**Require:** Training steps $T$; training set $X$; loss function $f$; state function $\chi$; discount factor: $\gamma$ ;
**Ensure:** Model parameters $w$, policy parameters $\theta$ of the actor network, and value parameters $\varphi$ of the critic network;
1: Initial parameters $\omega_0, \theta_0, \varphi_0$;
2: **for** $t = 0, ..., T$ **do**
3: Sample $x_i \in X, i \in 1, ..., N$ .
4: Extract state vector: $s_i^t = \chi(\omega^t, x_i)$.
5: //Actor network selects an action.
6: Computes learning rate $a_i^t = \pi_\theta(s_i^t)$.
7: //Update model parameters $\omega$.
8: Compute $\nabla f^t(x_i)$.
9: Update $\omega$: $\omega^{t+1} = \omega^t - a_i^t \nabla f^t(x_i)$.
10: //Update critic network by minimizing square error between estimation and label.
11: $r^t = f^t(x_i) - f^{t+1}(x_i)$
12: Extract state vector: $s_i^{t+1} = \chi(\omega^{t+1}, x_i)$
13: Compute $Q_\varphi(s_i^{t+1}, \pi_\theta(s_i^{t+1})), Q_\varphi(s_i^t, a_i^t)$
14: Compute $\delta^t$ according to Equation 7:
 $\delta^t = r^t + \gamma Q_\varphi(s_i^{t+1}, \pi_\theta(s_i^{t+1})) - Q_\varphi(s_i^t, a_i^t)$
15: Update $\varphi$ using the following gradients according to Equation 8 :
 $\nabla\varphi = \delta^t \nabla_\varphi Q_\varphi(s_i^t, a_i^t)$
16: // Update actor network
17: Sample $x_j \in X, j \in 1, ..., N, j \neq i$.
18: Extract state vector: $s_j^{t+1} = \chi(\omega^{t+1}, x_j)$.
19: Compute $a_j^{t+1} = \pi_\theta(s_j^{t+1})$.
20: Update $\theta$ from Equation 9:
 $\nabla\theta = \nabla_\theta \pi_\theta(s_j^{t+1}) \nabla_a Q_\varphi(s_j^{t+1}, a_j^{t+1})|_{a=\pi_\theta(s)}$
21: **end for**
22: **return** $\omega, \theta, \varphi$;

---

### 3.4 The Algorithm

The overall algorithm is shown in Algorithm 1. In each step, we sample an example (Line 3), extract the current state vector (Line 4), compute the learning rate using the actor network (Line 6), update the model (Lines 8-9), compute TD error (Lines 11-14), update the critic network (Line 15), and sample another example (Line 17) to update the actor network (Line 18-20). We would like to make some discussions about the algorithm.

First, in the current algorithm, for simplicity, we consider using only one example for model update. It is easy to generalize to a mini batch of random examples.

Second, one may notice that we use one example (e.g., $x_i$) for model and the critic network update, but a different example (e.g., $x_j$) for the actor network update. Doing so we can avoid that the algorithm will overfit on some (too) hard examples and can improve the generalization performance of the algorithm on the test set. Consider a hard example[2] in a classification task. Since such an example is difficult to be classified correctly, intuitively its gradient will be large and the learning rate given by the actor network at this step will also be large. In other words, this hard example will greatly change the model, while itself is not a good representative of its category and the learning algorithm should not pay much attention to it. If we feed the same example to both the actor network and the critic network, both of them will encourage the model to change a lot to fit the example, consequently resulting in oscillation of the training, as shown in our experiments. By feeding different examples to the actor and critic networks, it is very likely the critic network will find that the gradient direction of the example fed into the actor network is inconsistent with its own training example and thus criticize the large learning rate suggested by the actor network. More precisely, the update of $\omega$ is based on $x_i$ and the learning rate suggested by the actor network, while the training target of the actor network is to maximize the output of the critic network on $x_j$. If there is big gradient disagreement between $x_i$ and $x_j$, the update of $\omega$, which is affected by actor's decision, would cause the critic's output on $x_j$ to be small. To compensate this effect, the actor network is forced to predict a small learning rate for a too hard $x_i$ in this situation.

## 4 EXPERIMENTS

We conducted a set of experiments to test the performance of our learning rate learning algorithm and compared with several baseline methods. We report the experimental results in this section.

### 4.1 EXPERIMENTAL SETUP

We tested our method on two widely used image classification datasets: MNIST LeCun et al. (1998) and CIFAR-10 Krizhevsky & Hinton (2009). Convolutional neural networks (CNNs) are the standard model for image classification tasks in recent years, and thus the primary ML algorithm adopted the CNN model in all our experiments.

We specified our actor-critic algorithm in experiments as follows. Given that stochastic mini-batch training is a common practice in deep learning, the actor-critic algorithm also operated on minibatches, i.e., each step is a mini batch in our experiments. We defined the state $s^t = \chi(\omega^t, X_i)$ as the average loss of learning model $\omega^t$ on the input min-batch $X_i$. We specified the actor network as a two-layer long short-term memory (LSTM) network with 20 units in each layer, considering that a good learning rate for step $t$ depends on and correlates with the learning rates at previous steps while LSTM is well suited to model sequences with long-distance dependence. We used the absolute value activation function for the output layer of the LSTM to ensure a positive learning rate. The LSTM was unrolled for 20 steps during training. We specified the critic network as a simple neural network with one hidden layer and 10 hidden units. We use Adam with the default setting in TensorFlow optimizer toolbox Abadi et al. (2015) to train the actor and critic networks in all the experiments.

We compared our method with several mainstream SGD algorithms, including SGD, Adam Kingma & Ba (2014), Adagrad Duchi et al. (2011) and RMSprop Tieleman & Hinton (2012). For each of these algorithms and each dataset, we tried the following learning rates $10^{-4}, 10^{-3}, ..., 10^0$. We report the best performance of these algorithms over those learning rates. If an algorithm needs some other parameters to set, such as decay coefficients for Adam, we used the default setting in TensorFlow optimizer toolbox. For each benchmark and our proposed method, five independent runs are averaged and reported in all of the following experiments.

### 4.2 RESULTS ON MNIST

MNIST is a dataset for handwritten digit classification task. Each example in the dataset is a $28 \times 28$ black and white image containing a digit in $\{0, 1, \cdots, 9\}$. The CNN model used in the primary

---

[2]For example, an example may has an incorrect label because of the limited quality of labelers.

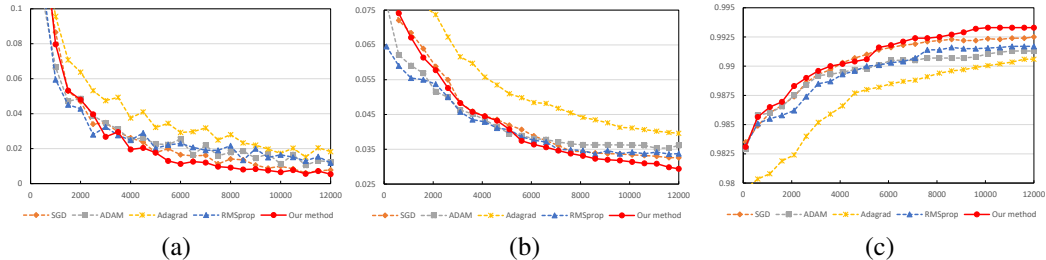

Figure 2: Results on MNIST. (a) Training loss. (b) Test loss. (c) Test accuracy. The x-axis is the number of mini batches.

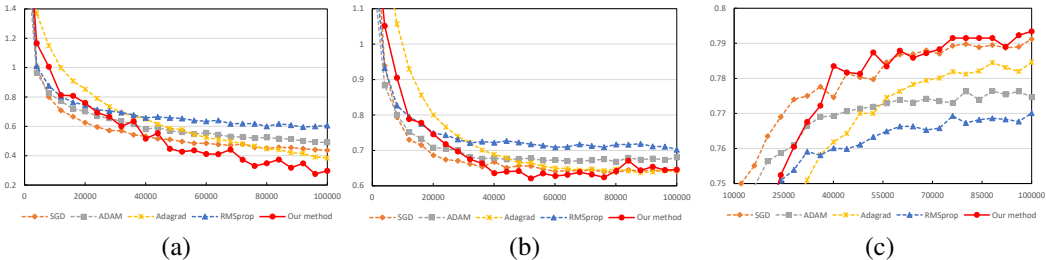

Figure 3: Results on CIFAR10. (a) Training loss. (b) Test loss. (c) Test accuracy. The x-axis is the number of mini batches.

ML algorithm is consist of two convolutional layers, each followed by a pooling layer, and finally a fully connected layer. The first convolutional layer filters each input image using 32 kernels of size $5 \times 5$. The max-pooling layer following the first convolutional layer is performed over $2 \times 2$ pixel windows, with stride 2. The second convolutional layer takes the outputs of the first max-pooling layer as inputs and filters them with 64 kernels of size $5 \times 5$. The max-pooling layer following the second convolutional layer is performed over $2 \times 2$ pixel windows, with stride 2. The outputs of second max pooling layer are fed to a fully connected layer with 512 neurons. Dropout was conducted on the fully connect layer with a dropout rate of 0.5. ReLU activation functions are used in the CNN model. There are 60,000 training images and 10,000 test images in this dataset. We scaled the pixel values to the [0,1] range before inputting to all the algorithms. Each mini batch contains 50 randomly sampled images.

Figure 2 shows the results of our actor-critic algorithm for learning rate learning and the baseline methods, including the curves of training loss, test loss, and test accuracy. The final accuracies of these methods are summarized in Table 1. We have the following observations.

- In terms of training loss, our algorithm has similar convergence speed to the baseline methods. One may expect that our algorithm should have significantly faster convergence speed considering that our algorithm learns both the learning rate and the CNN model while the baselines only learn the CNN model and choose the learning rates per some predefined rules. However, this is not correct. As discussed in Section 3.4, we carefully design the algorithm and feed different samples to the actor network and critic network. Doing so we can focus more on generalization performance than training loss: as shown in Figure 4, our algorithm achieves the best test accuracy.

Table 1: Error rate comparison on MNIST.

| Optimizer | Error Rate (%) |
| --- | --- |
| SGD | 0.75 |
| ADAM | 0.87 |
| Adagrad | 0.94 |
| RMSprop | 0.83 |
| Our method | **0.67** |

Table 2: Classification Accuracy on CIFAR-10.

| Optimizer | Accuracy |
| --- | --- |
| SGD | 78.74 |
| ADAM | 77.46 |
| Adagrad | 78.46 |
| RMSprop | 62.3 |
| Our method | **79.34** |

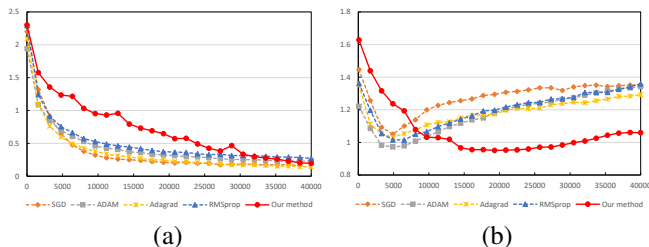
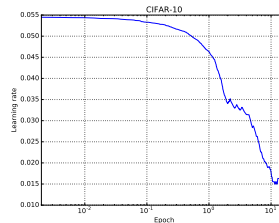

(a)　　　(b)

Figure 4: Results on CIFAR-10 with 20% training data. (a) Training loss. (b) Test loss.

Figure 5: The learning rate learned by actor network for CIFAR-10.

- Our algorithm achieves the lowest error rate on MNIST. Although the improvement looks small, we would like to point out that given that the accuracy of CNN is already close to 100%, it is a very difficult task to further improve accuracy, not to mention that we only changed learning rate policy without changing the CNN model.

## 4.3 RESULTS ON CIFAR-10

CIFAR-10 is a dataset consisting of 60000 natural $32 \times 32$ RGB images in 10 classes: 50,000 imagesfor training and 10,000 for test. We used a CNN with 2 convolutional layers (each followed by max-pooling layer) and 2 fully connected layers for this task. There is a max pooling layer which performed over $2 \times 2$ pixel windows, with stride 2 after each convolutional layer. All convolutional layers filter the input with 64 kernels of size $5 \times 5$. The outputs of the second pooling layer are fed to a fully connected layer with 384 neurons. The last fully connected layer has 192 neurons. Before inputting an image to the CNN, we subtracted the per-pixel mean computed over the training set from each image.

Figure 3 shows the results of all the algorithms on CIFAR-10, including the curves of training loss, the test loss and test accuracy. Table 2 shows the final test accuracy. We get similar observations as MNIST: our algorithm achieves similar convergence speed in terms of training loss and slightly better test accuracy than baselines. Figure 5 shows the learning rate learned by our method on CIFAR-10. To further understand the generalization performance of our algorithm, we ran all the

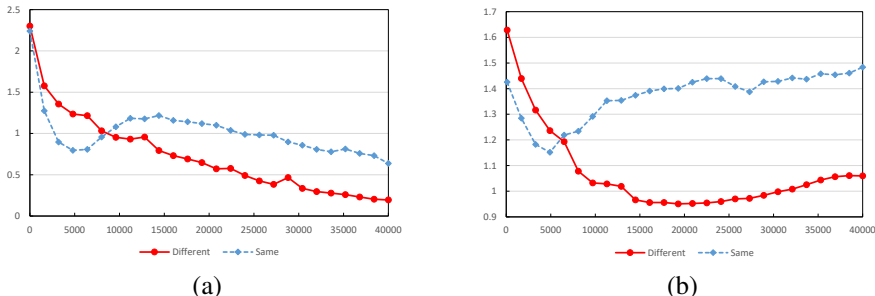

(a)　　　(b)

Figure 6: Results on CIFAR-10 with 20% training data. (a) Training loss. (b) Test loss. Our algorithm with $x_i = x_j$ is shown with blue line, and Our algorithm with $x_i \neq x_j$ is shown with orange line.

algorithms on two subsets of training data on CIFAR-10: one with only 20% training data The curves of training loss and test loss are shown in Figure 4. As can be seen from the figure, those baseline methods are easy to overfit and their test loss increases after 5000 steps (mini batches). In contrast, our algorithm is relatively robust and can prevent overfitting to some extent.

As we explained in Section 3.4, feeding different examples to the actor and critic networks is important to guarantee generalization ability. Here we conducted another experiment to verify our intuitive explanation. Figure 6 shows the results of two different implementations of our actor-critic algorithm on CIFAR-10. In the first implementation, we fed the sample examples to the two net-

Table 3: Error rate of different methods on different network architectures.

| Network | Methods | | | | | | | |
|---|---|---|---|---|---|---|---|---|
| | SGD | ADAM | Adagrad | RMSprop | vSGD-l | vSGD-b | vSGD-g | Our method |
| M0 | 7.60 | 8.70 | 7.52 | 10.91 | **7.50** | 7.89 | 8.20 | **7.50** |
| M1 | 2.34 | 4.12 | 2.70 | 6.17 | 2.42 | 2.44 | 4.14 | **2.04** |
| M2 | 2.15 | 3.85 | 2.34 | 3.81 | 2.16 | 2.05 | 3.65 | **2.03** |

works, i.e., $x_i = x_j$ in the algorithm, and in the second implementation, the input $x_j$ of the critic network is different from the input $x_i$ of the actor network. It is easy to see from the figure that setting $x_i = x_j$ tends to oscillate during training and leads to poor test performance. Thus, we need to feed different training data to the actor network and the critic network to ensure the performance of the algorithm.

### 4.4 COMPARISON WITH OTHER ADAPTIVE LEARNING RATE METHOD

We also compare our method with "vSGD" from previous by work Schaul et al. (2013), which can automatically adjust learning rates to minimize the expected error. This method tries to compute learning rate at each update by optimizing the expected loss after the next update according to the square norm of the expectation of the gradient, and the expectation of the square norm of the gradient. Note that our method learns to predict a learning rate at each time step by utilizing the long term reward predicted by a critic network.

For a fair comparison, we followed the experiments settings of Schaul et al. (2013), which designed three different network architectures for MNIST task to measure the performance. The first one is denoted by 'M0' which is simple softmax regression (i.e. a network with no hidden layer). The second one ('M1') is a fully connected multi-layer perceptron, with a single hidden layer. The third one (denoted 'M2') is a deep, fully connected multi-layer perceptron with two hidden layers. The vSGD has three variants in their paper. We referred to the results reported in their paper and compared our method with all of three variants of their algorithm (vSGD-l, vSGD-b, vSGD-g). The learning rates of SGD are decreased according to a human designed schedule, and the hyper-parameters of SGD, ADAM, Adagrad, RMSprop are carefully determined by their lowest test error among a set of hyper-parameters. All hyper-parameters can be found in Schaul et al. (2013).

The experimental results are reported in Table 3. It shows that our proposed method performs better than vSGD and other baseline methods, and is stable across different network architectures.

## 5 CONCLUSIONS AND FUTURE WORK

In this work, we have studied how to automatically learn learning rates for gradient based machine learning methods and proposed an actor-critic algorithm, inspired by the recent success of reinforcement learning. The experiments on two image classification datasets have shown that our method (1) has comparable convergence speed with expert-designed optimizer while achieving better test accuracy, and (2) can successfully adjust learning rate for different datasets and CNN model structures.

For the future work, we will explore the following directions. In this work, we have applied our algorithm to control the learning rates of SGD. We will apply to other variants of SGD methods. We have focused on learning a learning rate for all the model parameters. We will study how to learn an individual learning rate for each parameter. We have considered learning learning rates using RL techniques. We will consider learning other hyperparameters such as step-dependent dropout rates for deep neural networks.

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

# A   APPENDIX

A method of automatically controlling learning rate is proposed in the main body of the paper. The learning rate controller adjusts itself during training to control the learning rate. Here, we propose an improved version that can leverage experiences from several repeated training runs to learn a fixed learning rate controller. Empirically, this algorithm can achieve better performance than the previous one. Given that it requires more time for training the learning rate controller, this method is more suitable for training offline models.

In this algorithm, during every training run, we fix the actor network and compute the weighted sum of the gradients of its parameter $\theta$. The parameter is updated after each run (modified from Equation 9):

$$\nabla\theta = \Sigma_{t=1}^{T} h(t) \nabla_{\theta} \pi_{\theta}(s^{t+1}) \nabla_{a} Q_{\varphi}(s^{t+1}, a^{t+1})|_{a=\pi_{\theta}(s)}. \qquad (10)$$

$h(t)$ is weighted function which is used to amplify the feedback signal from the initial training stage. It is defined as $h(t) = 1/t$ in our experiments. An error rate of **0.48**% was achieved with 5 repeated training runs in MNIST experiment (the same setting as Table 1), and in CIFAR-10 experiment (the same setting as Table 2), **80.23%** accuracy was achieved with 10 training runs. This method showed better performance in both experiments.

