# Peer review of "An Actor-critic Algorithm for Learning Rate Learning"

_ICLR 2017 — rejected_

[Official Review · AnonReviewer3 · rating 4 · confidence 4 · 15 Dec 2016]
**No comparisons to recent alternatives**

The paper proposes using an actor-critic RL algorithm for training learning rate controllers for supervised learning. The proposed method outperforms standard optimizers like SGD, ADAM and RMSprop in experiments conducted on MNIST and CIFAR 10.

I have two main concerns. One is the lack of comparisons to similar recently proposed methods - "Learning Step Size Controllers for Robust Neural Network Training" by Daniel et al. and "Learning to learn by gradient descent by gradient descent" by Andrychowicz et al. The work of Daniel et al. is quite similar because it also proposes using a policy search RL method (REPS) and it is not clear what the downsides of their approach are. Their work does use more prior knowledge as the authors stated, but why is this a bad thing?

My second concern is with the experiments. Some of the numbers reported for the other methods are surprisingly low. For example, why is RMSprop so bad in Table 2 and Table 3? These results suggest that the methods are not being tuned properly, which reinforces the need for comparisons on standard architectures with previously reported results. For example, if the baselines used a better architecture like a ResNet or, for simplicty, Network in Network from this list:

[Official Review · AnonReviewer1 · rating 3 · confidence 5 · 15 Dec 2016 (modified: 23 Jan 2017)]

In the question response the authors mention and compare other works such as "Learning to Learn by Gradient Descent by Gradient Descent", but the goal of current work and that work is quite different. That work is a new form of optimization algorithm which is not the case here. And bayesian hyper-parameter optimization methods aim for multiple hyper-parameters but this work only tune one hyper-parameter.
The network architecture used for the experiments on CIFAR-10 is quite outdated and the performances are much poorer than any work that has published in last few years. So the comparison are not valid here, as if the paper claim the advantage of their method, they should use the state of the art network architecture and see if their claim still holds in that setting too.
As discussed before, the extra cost of hyper-parameter optimizers are only justified if the method could push the SOTA results in multiple modern datasets.
In summary, the general idea of having an actor-critic network as a meta-learner is an interesting idea. But the particular application proposed here does not seems to have any practical value and the reported results are very limited and it's hard to draw any conclusion about the effectiveness of the method.

[Official Review · AnonReviewer2 · rating 5 · confidence 4 · 20 Dec 2016]
**Interesting application of actor-critic methods, but difficult to assess relative to other adaptive learning algorithms**

The authors present a method for adaptively setting the step size for SGD by treating the learning rate as an action in an MDP whose reward is the change in loss function. The method is presented against popular adaptive first-order methods for training deep networks (Adagrad, Adam, RMSProp, etc). The results are interesting but difficult to assess in a true apples-to-apples manner. Some specific comments:

-What is the computational overhead of the actor-critic algorithm relative to other algorithms? No plots with the wall-time of optimization are presented, even though the success of methods like Adagrad was due to their wall-time performance, not the number of iterations.
-Why was only a single learning rate learned? To accurately compare against other popular first order methods, why not train a separate RL model for each parameter, similar to how popular first-order methods adaptively change the learning rate for each parameter.
-Since learning is a non-stationary process, while RL algorithms assume a stationary environment, why should we expect an RL algorithm to work for learning a learning rate?
-In figure 6, how does the proposed method compare to something like early stopping? It may be that the actor-critic method is overfitting less simply because it is worse at optimization.

[Final Decision · Program Chairs · 06 Feb 2017]
**ICLR committee final decision**

The authors use actor-critic reinforcement learning to adjust the step size of a supervised learning algorithm. There are no comparisons made to other, similar approaches, and the baselines are suspiciously weak, making the proposed method difficult to justify.